# Is it that simple? The use of linear models in cognitive neuroscience
*CCN Generative Adversarial Collaboration Proposal*

## Scientific question

Do linear models provide an accurate, interpretable, and biologically plausible description of brain activity?

## Background

Modern cognitive neuroscience heavily relies on linear models. Such models are used to map between patterns of brain activity and a measure X, where X can be a feature/function of the stimulus (1–10), a behavioral measure (e.g., 11–13), or even brain activity in another species (14,15). The use of linear (as opposed to nonlinear) models is widespread for two main reasons: (a) linear readout is considered to be neurally plausible and thus informative of the underlying neural representations (16–19), and (b) linear models are relatively easy to build and can generalize successfully even in small data regimes (17,20,21).

In recent years, increased availability of large datasets and computational resources has enabled researchers to overcome some of the practical limitations of nonlinear approaches. As a result, many prediction-oriented neuroscience studies have begun to apply nonlinear models to identify neural correlates of brain disorders (22–25) or behavioral traits (26–28). However, basic cognitive neuroscience remains firmly grounded in linear models, resulting in a gap between prediction-oriented and explanation-oriented approaches.

## The Controversy

Although the linear readout assumption is widely accepted in cognitive neuroscience, neural computations are, in fact, often nonlinear (29–35). Further, even if we accept the linear readout assumption at the level of individual neurons, it might not hold for signals recorded from outside the skull (36–38) or signals based on indirect measures of neural activity, such as blood flow (39,40). Finally, even if the transmission of signals from one step of a neural computation to another can be approximated with a linear transform, the linearity assumption might break down once we consider multiple successive computations (41,42) or activity aggregated across large neural populations (43–45). As a result, some recent neuroimaging studies have advocated the use of nonlinear models, arguing that they represent a more plausible view of neural interactions within and between brain regions (46–49), at least for higher-level associative cortex (50–52).

The empirical success of linear models seems to speak to the usefulness of the linear readout assumption. However, the persistent focus on linear transformations may be stalling the field, and a shift toward nonlinear models may yield important insights about brain function. We therefore propose to combine

theoretical and experimental approaches in order to examine the validity, benefits, and limitations of linear vs. nonlinear models applied to neuroimaging data.

## Competing Hypotheses

Hypothesis 1: Linear models provide an accurate, interpretable, and biologically plausible interpretation of brain activity.

Hypothesis 2: Nonlinear models provide an accurate, interpretable, and biologically plausible interpretation of brain activity, which cannot be achieved with linear models alone.

## Approach

We propose an integrated, two-pronged approach for evaluating the use of linear and nonlinear models in cognitive neuroscience. First, we will synthesize existing literature on the linear readout assumption and introduce novel information-theoretic approaches for model evaluation. Then, we will build upon those theoretical insights to evaluate the models' performance on existing datasets.

1. The Theory Branch
   a. Establish the theoretical validity of the linear readout assumptions when applied to neuroimaging data.
   b. Develop information-theoretic criteria for evaluating the use of linear and nonlinear models in neuroimaging research (see 53–55).
2. The Empirical Branch
   a. Evaluate practical limitations of linear vs. nonlinear models, such as the amount of data required for successful performance and the upper limit on feature complexity (see 56–58).
   b. Integrate theory-driven and practical considerations to develop goals and metrics enabling a systematic comparison of linear vs. nonlinear model performance.
   c. Compare the predictive and explanatory power of linear vs. nonlinear models when applied to three different domains:
      i. Mapping from stimulus features to neural activity.
      ii. Mapping from neural activity to behavior.
      iii. Mapping from neural activity in one brain region to neural activity in another brain region.

## Concrete outcomes

1. An information-theoretic framework for the use of linear vs. nonlinear models with neuroimaging data.
2. A detailed set of guidelines for the use of linear vs. nonlinear models with neuroimaging data, based on both theoretical and empirical considerations.
3. An online platform enabling researchers to systematically compare linear and nonlinear models according to a predefined set of metrics (see, e.g., 8).

## Benefit to the community

Given the overwhelming use of linear models in the field, we believe that a thorough examination of the linear readout assumption is required to ensure that researchers do not overlook important insights about the brain by unnecessarily restricting the set of models they consider. On the other hand, given the potentially unbounded complexity of nonlinear models, neuroscientists must be careful in their application and interpretation. If we demonstrate the benefit of nonlinear models, at least in some cases, our work may catalyze a new line of inquiry in cognitive computational neuroscience. If we demonstrate that linear models satisfy the field's criteria of being accurate, interpretable, and biologically plausible, our work will allow future researchers to continue relying on linear rather than nonlinear approaches, thus saving money, time, and computational resources.

Thus, we expect that our findings will be relevant to any neuroscientist who uses multivariate methods to analyze neuroimaging data. They also have the potential to benefit other researchers investigating complex information processing systems (e.g. artificial neural networks).

## Core members

| Team linear models: | Martin Schrimpf (graduate student, MIT) |
| | Leyla Isik (assistant professor, Johns Hopkins University) |
| Team nonlinear models: | Anna Ivanova (graduate student, MIT) |
| | Stefano Anzellotti (assistant professor, Boston College) |
| The advisory team: | Noga Zaslavsky (postdoctoral fellow, MIT) |
| | Evelina Fedorenko (associate professor, MIT) |

## Member roles

All members will contribute to organizing the workshop and writing the paper. In addition, we will perform the following tasks:

| Anna Ivanova: | examine the theoretical assumptions underlying linear models applied to neuroimaging data. |
| Martin Schrimpf: | evaluate linear and nonlinear model performance on existing neuroimaging datasets. |
| Leyla Isik: | develop goals/metrics to systematically evaluate linear vs. nonlinear model performance. |
| Stefano Anzellotti: | examine methodological benefits and limitations of linear vs. nonlinear models. |
| Noga Zaslavsky: | develop information-theoretic methods for studying and evaluating linear and non-linear models; guide the theory branch of the project. |
| Evelina Fedorenko: | guide the empirical branch of the project and the integration of final results. |

**Signed:** Anna Ivanova, Martin Schrimpf, Leyla Isik, Stefano Anzellotti, Noga Zaslavsky, and Evelina Fedorenko

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
