# OpenReview forum: "Is it that simple? The use of linear models in cognitive neuroscience"
_ccneuro.org/CCN/2020/Workshop/GAC_

### Official Review · ~RT_Pramod1 · 2020-08-26
**Not that simple after all!**

**Rating:** 8
**Soundness:** Strongly agree
**Confidence:** 5

**Review:**

<p>Here are my comments on the proposal “Is it that simple? The use of linear models in cognitive neuroscience”. The proposal starts by asking if linear models provide accurate, interpretable, and biologically plausible description of brain activity. If we, for this argument, consider that the end goal of any model building enterprise is to make better predictions on unseen data (quantified using a suitable metric), then it really doesn’t matter if we have a linear or a non-linear model as long as it has the best predictions. However, we also want the models to be interpretable so that we can relate to and extend theories and hypotheses in cognitive neuroscience that have been traditionally ‘word-based’. In addition, using either model class also depends on the problem (prior knowledge of the system and dataset) at hand. Below, I will try to highlight some issues in fitting linear and non-linear models --
<li>Once we fix the variables (or features), there are usually only a few choices to be made for a linear model. However, in case of non-linear models, the possibilities are innumerable requiring additional knowledge of the system under consideration to constrain the model. </li>
<li>The kind of models we can build is largely dictated by the nature of our experiments and the amount of data we have. For example, linear models could be useful while predicting behavior from neural activity in the higher visual cortex whereas it might not be suitable for predicting the animal’s behavior from activity in the primary visual cortex. Further, linear models might be more suitable when we have less data to constrain our models. However, this might become a non-issue in the era of large datasets in neuroscience. </li>
<li>In many cases, we can linearize a model by suitably transforming the variables (similar to the kernel trick used in machine learning). Again, this assumes that we know what the non-linearities are (same as above) but in some cases is tractable (using deep convolutional neural networks to transform pixels to features [1, 2, 3]; semantic embedding [4] etc). In some cases, interactions between variables (modeled using non-linearities) might suggest the existence of other underlying variable(s) that when taken together with the original variables can lead to linear models [5]. </li>
In light of the above points, I think throwing out linear models for want of less ‘simple’ models is too drastic of a step! In fact, one might rule in favor of either hypothesis depending on what they’re interested in – non-linear models might be the better predictors whereas linear models might lead to more interpretable results, on the same dataset! As outlined in the proposal, establishing theories and sensitive metrics to evaluate various models will be a useful contribution to the field. However, all of these will be useful only when used properly in the context of the scientific experiment, the quality of the data collected and the eventual interpretability of the results. In addition, the workshop should also consider how to include behavioral data as a constraint to reduce the space of models (as we can think of scenarios where a particular variable can be read-out from neural activity but turns out to be behaviorally irrelevant!). Overall, I think this is an excellent proposal for a GAC workshop and am excited to hear what the field thinks about this issue and specifically about how non-linear models can reveal hitherto unknown things about the brain.</p>

References:
[1] Yamins, D. L., & DiCarlo, J. J. (2016). Using goal-driven deep learning models to understand sensory cortex. Nature neuroscience, 19(3), 356-365.
[2] Khaligh-Razavi, S. M., & Kriegeskorte, N. (2014). Deep supervised, but not unsupervised, models may explain IT cortical representation. PLoS computational biology, 10(11), e1003915.
[3] Pramod, R. T., & Arun, S. P. (2016). Do computational models differ systematically from human object perception?. In Proceedings of the IEEE Conference on Computer Vision and Pattern Recognition (pp. 1601-1609).
[4] Huth, A. G., De Heer, W. A., Griffiths, T. L., Theunissen, F. E., & Gallant, J. L. (2016). Natural speech reveals the semantic maps that tile human cerebral cortex. Nature, 532(7600), 453-458.
[5] Pramod, R. T., & Arun, S. P. (2014). Features in visual search combine linearly. Journal of vision, 14(4), 6-6.

**Comments:**

<p>The proposal starts by asking if linear models provide accurate, interpretable, and biologically plausible description of brain activity. If we, for this argument, consider that the end goal of any model building enterprise is to make better predictions on unseen data (quantified using a suitable metric), then it really doesn’t matter if we have a linear or a non-linear model as long as it has the best predictions. However, we also want the models to be interpretable so that we can relate to and extend theories and hypotheses in cognitive neuroscience that have been traditionally ‘word-based’. In fact, one might rule in favor of either hypothesis depending on what they’re interested in – non-linear models might be the better predictors whereas linear models might lead to more interpretable results, on the same dataset! As outlined in the proposal, establishing theories and sensitive metrics to evaluate various models will be a useful contribution to the field. However, all of these will be useful only when used properly in the context of the scientific experiment, the quality of the data collected and the eventual interpretability of the results. In addition, the workshop should also consider how to include behavioral data as a constraint to reduce the space of models (as we can think of scenarios where a particular variable can be read-out from neural activity but turns out to be behaviorally irrelevant!). Overall, I think this is an excellent proposal for a GAC workshop and am excited to hear what the field thinks about this issue and specifically about how non-linear models can reveal hitherto unknown things about the brain.</p>

**Controversy:**

Agree

**Definition:**

Agree

**Expertise:**

Strongly agree

**Outcomes:**

Agree

---

### Official Review · ~Nikhil_Parthasarathy1 · 2020-08-26
**great topic and mostly sound proposal, but hypotheses are possibly oversimplified given the complexity of the issues**

**Rating:** 7
**Soundness:** Strongly agree
**Confidence:** 4

**Review:**

This proposal seeks to evaluate the use of linear and nonlinear models to interpret/decode from patterns of brain activity. Current methods in cognitive science primarily rely on linear models (primarily due to their simplicity and interpretability), and the authors would like to question whether this is appropriate and whether nonlinear methods might in fact provide more accurate and possibly still interpretable models of brain activity. The authors plan to evaluate linear vs. nonlinear models from both a theoretical and experimental perspective.

**Originality and Significance:** I believe this topic is highly relevant today and important for the community to discuss because of the proliferation of projects now that center on recording brain activity (in various modalities) and attempting to build encoding/decoding models from stimulus to brain activity to behavior. Developing clarity on how we should think about linear vs. nonlinear models in these contexts is extremely valuable.

**Quality / Clarity**:
**Pros**: The high-level topic of discussion is great. Additionally, the proposal is well-written and easy to follow. I also like the proposed two-pronged approach to understanding this issue both using theory to understand when linear readouts are optimal or close to optimal and determining information-theoretic criteria fro evaluating linear vs. nonlinear models. The empirical approach seems sound and thorough (but see critiques below).
**Critiques and Questions**: I have a few critiques regarding the clarity of the proposal and the oversimplification of the issues at hand:
1) I have an issue with the way in which the hypotheses are stated in that they are not necessarily contradictory and they fail to capture the nuances of the problem effectively. Each hypothesis states either linear or nonlinear models "provide an accurate, interpretable, and biologically plausible interpretation of brain activity". The authors make it seem that these three criteria: accuracy, interpretability, and biological plausibility should be weighted equally across use-cases, when in reality it is the *use-case of the model that determines which of these criteria are most important.*  I would encourage the authors to not overgeneralize the issue and evaluate all linear/nonlinear models by all of these criteria; rather, I would suggest assuming various research questions and in each question assessing the power of nonlinear vs linear models given the criteria that are most important for that research goal. For example, if we are using a decoder to establish limits on what information about a stimulus can be read-out or reconstructed from a neural population then we might want a powerful nonlinear decoder that is simply optimized for maximum accuracy, even if it is not interpretable. However, if we are attempting to understand how the brain actually reads out a specific decision from neural activity then biological plausibility and interpretability may matter a lot more even if the model is less accurate.
2) Given the previous concern, it is also unclear to me what problems the authors will focus on and how this will impact the analyses. They mention data at the level of individual neurons, EEG based activity, fMRI activity etc. and mention 3 separate domains (encoding models for stimulus to neural activity, decoding models from neural activity to behavior, and models for transformations between brain regions). To me this seems like far too many contexts to do a thorough analysis of the benefits and limitations of multiple models because each context and data source will again influence the criteria that matter for evaluation of the models.
3) The authors say they would like to compare the power of linear vs. nonlinear models in three domains. With regards to the first: "Mapping from stimulus features to neural activity", I'm not sure what controversy they see here. Generally, (at least in visual neuroscience) nonlinear encoding models have been standard for modeling both early visual areas (i.e. Vnitch et al. 2015) all the way to higher visual areas (Yamins et al. 2014) and while there is debate over how to get more interpretable models, I dont think there is much controversy over whether linear models are sufficient.  What does seem to be more controversial are the second  and third domains because of the use of *linear readouts* from neural representations to decode other regions of brain activity, stimulus features, or behavior. However, in a sense this is just because the "nonlinear" piece has been built into the encoding and so we assume that a linear readout will be effective and interpretable. I do agree that it is totally valid to question this assumption for linear decoding, but I wonder if the authors could clarify if this is the primary controversy because I think most people would agree that nonlinearities have to play a role in the entire transformation from stimulus to behavior. if this is the primary concern then I would suggest focusing on domains 2 and 3 to ground the debate.
4) Finally, the authors claim they will evaluate the models on interpretability and biological plausibility, but it seems to me that these criteria are dependent on the level of abstraction that you care about (i.e. explaining single cells or decomposing decisionmaking at the population level etc.) I just wonder if this will be hard to evaluate in the "benchmark" system that is being proposed.

**Comments:**

I think the proposal is strong in it's significance and conceptualization and I believe there is a lot to be learned through the proposed analyses. My main concerns (see above) have to do with being more specific with the controversies and possibly isolating fewer domains to look at because it seems like the proposed work could be too overgeneralized given the stated goals. However, I think the proposed theory, experiment, and benchmarking will be very valuable to the community and could bring in a lot of great discussion about when it is appropriate and desired to use linear vs. nonlinear models in neuroscience and cognitive science.

**Controversy:**

Agree

**Definition:**

Agree

**Expertise:**

Strongly agree

**Outcomes:**

Agree

---

### Official Review · ~Melikasadat_Emami1 · 2020-08-26
**Is it really linear vs. nonlinear?**

**Rating:** 7
**Soundness:** Agree
**Confidence:** 4

**Review:**

This is a timely and impactful proposal in both theoretical and practical sense. I think the question is interesting but very general and it might not be that simple to answer. It would be great to frame it differently using the knowledge of the task and dataset and some of the major factors involved. I believe the results would benefit the community.


**Comments:**

Comments:
I think one of the most important things to consider is to determine if we want better predictability or interpretability. For the latter, the bias is towards the linear models although linearity does not necessarily entail interpretability. For prediction, on the other hand, non-linear models have shown to be able to provide better results than the linear models. Recent advances in machine learning have shown that very complex, over-parametrized models can perfectly fit data even with random labels. I think we need to first understand which one of these is more important to us depending on the task.
Then, for that specific goal (more predictable vs more interpretable model), I think what would be really great to understand is the cases where we can gain more from using non-linear models. I think it is almost impossible to find some neuroimaging data that non-linear models cannot add anything to what can be achieved from linear models. Depending on the size of the dataset, the specific task, the brain region where the data is recorded from, and special structures in the data, etc, we want to understand how much we can gain from non-linear models. I think the "linear vs nonlinear" is a general question considering the number of factors involved. It might be better if we try to understand what are the conditions that non-linear models are more useful and consider different tasks separately.
Another interesting question is the possibility of designing a model, based on the aforementioned factors, such that we gain even more (on predictability or interpretability) from nonlinear models (see [1]). For example, using a low-rank structure in a nonlinear decoder can significantly increase its performance compared to other linear/nonlinear models for a classification task with ECoG data. This might seem a difficult question, but exploring the possible directions to answer it would be beneficial.

[1] Emami, M., Sahraee-Ardakan, M., Pandit, P., Fletcher, A. K., Rangan, S., Trumpis, M., ... & Viventi, J. Low-Rank Nonlinear Decoding of $\mu $-ECoG from the Primary Auditory Cortex. arXiv preprint arXiv:2005.05053.

**Controversy:**

Agree

**Definition:**

Agree

**Expertise:**

Agree

**Outcomes:**

Agree

---

### Public Comment · ~Doby_Rahnev1 · 2020-08-11
**Are the two hypotheses mutually exclusive?**

I find myself in agreement with both Hypotheses 1 and 2, so they don't seem to be mutually exclusive. Maybe something more is implied in one of both of the hypotheses? Is the idea behind Hypothesis 1 that linear models are the ONLY ones that satisfy these criteria?

---

### Public Comment · ~Marc_Schwartz-Palleja1 · 2020-08-14
**One model fits all?**

Regarding to your hypothesis, are you trying to proof either one or the other? When choosing a model, the research question should guide your model selection process. Linear models might prove more than sufficient for a particular question. I would just add or comment that the decision process to select a model depends more on the question you want to ask and which level of explanation you are looking for, as in descriptive, mechanistic or normative.

Nonlinear models do indeed need a more thorough inspection and I find this presented idea excellent!

---

### Public Comment · ~Laura_Gwilliams1 · 2020-08-17
**We don't know what we're missing**

Linear models are known to be imperfect approximations of neural activity. However, using linear models to analyse neural data is usually considered "good enough" to provide evidence for representation X, or to show that theoretical account Y better explains neural responses than theoretical account Z. The success of these models has thus left the underlying linearity assumption aside, often undiscussed or unconsidered.

The current proposal brings this linearity assumption to direct attention. It tests the extent to which assuming a linear relationship between neural activity and a regressor of interest (e.g. stimulus features, behaviour, other neural activity) is truly a problem. And further, in what specific situations adding non-linearities provides superior explanatory power as compared to linear equivalents.

The true power of this proposal is in understanding the extent of the problem. It is possible that the additional complications of using non-linear models (e.g. additional training data, complications in interpretability) is simply not worth it. That non-linear models do not provide a substantial gain in explanatory power. It is also possible, however, that certain relationships between neural activity and e.g. stimulus features are undetectable using linear techniques, and can only be recovered using non-linear modelling. This line of work promises to theoretically and empirically describe the costs and advantages of using non-linear modelling, providing principled and motivated steps forward for the field as a whole.

---

### Public Comment · ~Martin_N_Hebart1 · 2020-08-17
**Go team linear!**

I was asked to provide feedback on the proposal of this Generative Adversarial Collaboration, and I'm more than happy to do so.

This is an exciting and important topic: linear models are used almost ubiquitously in neuroscience (specifically cognitive neuroscience), while non-linear models are common mostly in machine learning research or predictive modeling. Why not, then, adopt non-linear models more widely?

While I thought the arguments were set up very nicely and while I enjoyed reading the proposal, I think one important reason in favor of using linear models is barely discussed: interpretability. Interpretability is an important feature for many (not all!) linear models: in many cases, the weights of a general linear model can be interpreted as the contribution of the model variable to measured signal, and even a linear decoding model can be converted appropriately to make weights interpretable (Haufe et al., 2014, Neuroimage). This makes it possible for us to understand what is going on in the brain, for example what stimuli a voxel prefers, what allows our classifier to discriminate two patterns of brain activity, or even how the similarity of brain activity patterns can be related to each other. Highly non-linear models, on the other hand, don't easily offer this level of transparency and make it more difficult to move from mere prediction to interpretation.

Let's consider a thought experiment. Assume I have a neuroscience method that allows me to access all relevant information from retinal ganglion cells when presenting different stimuli to human observers. I then apply a highly non-linear model to the measured activation patterns. As a result, the model now allows me to tell what the stimulus is, allows me to generalize from one object size, orientation, or position to another, and even lets me figure out what purchasing actions are the most appropriate to take for valuable objects given my budget. The question that bugs me: What did we learn in these examples about the brain? If our goal is not only to build brain-machine interfaces, how can we move from prediction to interpretation (Hebart & Baker, 2018, Neuroimage)?

Of course, there are cases where prediction alone will allow for meaningful interpretation, which has been the case for much of multivariate pattern analysis research, and examples of which we see on a regular basis. However, one has to be very careful not to *induce* the results one would want to find by the use of a non-linear model. In addition, it has been shown repeatedly that even for linear decoding, not all patterns are read out in behavior (Williams et al., 2007, Nature Neuroscience; Ritchie et al., 2015, PLoS Computational Biology). How much worse is it then for non-linear methods?

How, then, can one get the best of both worlds? One approach we can choose is to *linearize* non-linear components of a model. This process makes explicit what computations we assume to precede the current processing stage. This can be applied both for known functional relationships, such as population receptive fields, but also for less transparent transition functions, such as deep convolutional neural networks (van Gerven, 2017, Journal of Mathematical Psychology) or semantic embeddings (Huth et al., 2016, Nature). In other words: Linearization allows us to use non-linear models while keeping them in check.

At the same time, this approach may impose limitations that are unnecessarily strict for a range of scientific questions. Working out what scientific questions can be addressed specifically with non-linear models while remaining in the domain of biological plausibility is an important avenue for this GAC, and I am looking forward to the ensuing discussions and results of this proposal.

---

### Public Comment · ~Christopher_Baldassano1 · 2020-08-20
**Great proposal with both theoretical and practical impact!**

This is a great topic for a GAC, since understanding the pros and cons of nonlinear models is of immediate practical importance (what kinds of models should researchers be using) and also tells us something fundamental about how information is represented in the brain. The proposed members all have relevant expertise in this area, and have identified some domains in which linear or nonlinear models show superior performance.

I do think that Hypothesis 2 is hard to falsify - showing that there is *no* situation in which nonlinear models provide additional predictive/explanatory power is probably impossible. It might be better to frame this as understanding how linear vs. nonlinear performance depends on:

-dataset size (number of neural/stimulus features, number of timepoints/stimuli)

-brain region

-data modality (electrophysiology, EEG, MEG, fMRI...)

-learning algorithm (e.g. recent ML results have shown that very complex models can generalize well if trained via SGD)

-model complexity/regularization

-stimulus complexity

The goal should be to understand the parts of this space in which nonlinear models are useful, i.e. identify the conditions under which Hypothesis 2 is true.

---

### Public Comment · ~Pouya_Bashivan1 · 2020-08-20
**Is there a silver model?**


This proposal discusses an important point on the sufficiency of linear models to make inferences based on observed brain activity.
In addition to linear and nonlinear models, I would also include another class of models widely used in neuroscience that corresponds to single-feature comparisons (e.g. neural tunings). Having these three classes of models in mind, one can identify a spectrum of possible models that includes one-to-one comparison methods  (i.e. single-feature comparison) on one end and complex non-linear models (e.g. artificial neural networks with nonlinearities) on the other.

Going back to the proposal, selecting the right type of model is important but I’m afraid there will not be a silver bullet (silver model class) and the choice will ultimately depend on a number of factors including the goal of the analysis and possible regularities in the representations.

For illustration, consider the following examples regarding the dependence of model choice on the goal of the study.
* If the goal is to show whether neurons in a certain area are sensitive to a specific feature, using one-to-one comparison methods might be as good as (or better than) more complex methods. E.g. orientation tuning in V1 cortex.
* If the goal is to assess the similarity of a feature space to an ensemble of neural recordings from the brain, again simpler methods would provide better judgement since, the more complex the mapping is the more it can artificially improve the similarity score. E.g. comparing layer activations in Alexnet and recordings from IT cortex.
* If the goal is to build a maximally predictive model to be used in a potential application, then all we should care about is achieving the best similarity score. In this case it makes sense to use the most complex model we can come up with as long as its predictions still generalize.

---

### Public Comment · ~Mariya_K_Toneva1 · 2020-08-21
**Potential for relation to current debate in NLP about probe complexity**

The question of linear vs non-linear encoding/decoding models is interesting and it investigates an often-made assumption in computational neuroscience that linear models are good enough. The addition of a theoretical investigation makes this even more exciting.

There are a few questions/suggestions that come up from the proposal:
- Will the empirical branch investigate different types of brain recordings? It is plausible that a linear encoding/decoding model is a better approximation of the relationship to one type of brain recording (let’s say fMRI which is smoothly varying), but not for other brain recordings (MEG, EEG, ECoG).
- Will the investigation be restricted to one domain (e.g. vision, language), or are the findings expected to be universal?
- There is a closely related recent debate in NLP about “probe” complexity, where a probe is a supervised model trained to extract specific linguistic information about text (e.g. part of speech, dependency parse, etc.) from the corresponding deep neural network feature representations. In these terms, an encoding/decoding model (that relates a feature representation of text to the corresponding brain recordings) can be thought of as a type of a probe. In NLP, researchers use both linear and non-linear probes, and there is an on-going discussion about probe complexity, including a recent discussion based on information theory (Pimentel et al. 2020 ACL, https://arxiv.org/abs/2004.03061). Brain recordings and linguistic labels are two very different kinds of signals, but it may be worthwhile to examine the similarities in case some of the NLP findings about probes can be applied to encoding/decoding models.

---

### Public Comment · ~Jean-Remi_King1 · 2020-08-23
**Interpretation vs Prediction > Linear vs Nonlinear**

I was asked to provide feedback to the present proposal, and am thus happy to forward my comments here.

The power and limits of linear models is an important and timely topic in cognitive and computational neurosciences.

There has been an ongoing discussion on this issue across multiple labs (e.g. [1-3]). A summary of the main arguments can be found in a chapter of the last edition of Cognitive Neuroscience book [4].

I have three minor and one major comments to the present proposal, which I hope will help making the debate successful:

Minor:
- The current scope may be too wide: e.g. the current references suggest that the discussion will cover the use of nonlinear models for Alzheimer and autism diagnoses, behavioral analyses and fMRI representations. Focusing on the specific issue of neural representations / encoding, and whether the linearity assumption is good/useful *there*, would probably provide a clearer benefit to the community than a broad discussion.

- I would recommend anchoring the debate in a clearly delimited set of paradigms: e.g. (1) object recognition, (2) speech comprehension and (3) spatial navigation. Such paradigms are sufficiently different to ensure that most CCN communities would understand that the questions at stake are general, while still providing clear conclusions and limitations for each argument.

- It may be important to discriminate "static" (e.g. BOLD) and "dynamic" representations (e.g. ecog, spikes etc). When the brain signals are well resolved in time, it is common to use non-linear dynamical system to model what they do. At the extreme, modeling a spike from synaptic input is challenging a strictly linear model.

Major:
- I believe it is important to more clearly separate the issue of prediction vs interpretation from the issue of linear vs nonlinear models.

   i) linearity does not entail interpretability: For example, in Huth, et al's paper [5], the authors used a linear model to map word embedding onto BOLD fMRI responses. However, when the authors attempted to interpret the coefficients of this encoding model, they appeared to be challenged: only the first 2 principal components could be clearly interpreted, while the ~20 other could not. This is an example where a high-dimensional linear model is arguable not easily interpretable - but perhaps the authors would argue otherwise.

   ii) Vice-versa, when simple enough, nonlinear models can be interpretable, in that we can internalize how they behave and what they would predict: e.g. Lorenz system and Schrödinger's nonlinear equations can be arguably interpreted under that definition (*).

In this view, I would argue that the debate on the use of (non)linear models should be relatively simple to address. The much deeper epistemological challenge is whether we, as scientist, should focus on interpretation or on prediction. The lessons that I personally take from other disciplines, and from quantum mechanics and deep learning in particular, make me lean towards the latter. But I would be happy to hear the counter-arguments.

References:

[1] Benjamin, Ari S., et al. "Modern machine learning far outperforms GLMs at predicting spikes." bioRxiv (2017): 111450.

[2] Richards, B. A., Lillicrap, T. P., Beaudoin, P., Bengio, Y., Bogacz, R., Christensen, A., ... & Gillon, C. J. (2019). A deep learning framework for neuroscience. Nature neuroscience, 22(11), 1761-1770.

[3] Naselaris, T., Kay, K. N., Nishimoto, S., & Gallant, J. L. (2011). Encoding and decoding in fMRI. Neuroimage, 56(2), 400-410.

[4] King, J. R., Gwilliams, L., Holdgraf, C., Sassenhagen, J., Barachant, A., Engemann, D., ... & Gramfort, A. (2018). Encoding and decoding neuronal dynamics: Methodological framework to uncover the algorithms of cognition. Cognitive Neuroscience VI, Poeppel, Mangun & Gazzaniga

[5] Huth, A. G., De Heer, W. A., Griffiths, T. L., Theunissen, F. E., & Gallant, J. L. (2016). Natural speech reveals the semantic maps that tile human cerebral cortex. Nature, 532(7600), 453-458.

(*) edit: change e=mc^2 example into other famous physics equation, as c^2 is a constant.

---

### Public Comment · ~Katharina_Dobs1 · 2020-08-25
**(When) are linear models sufficient?**

I enjoyed reading the proposal. The question is interesting and timely given that more and more nonlinear models are being applied to cognitive neuroscience. My main comment is that the outcome might be continuous rather than categorical such that nonlinear methods might be beneficial for some data/problems/applications but not for others. Testing all these different settings is clearly out of the scope of this proposal. However, it could be beneficial to consider different types of datasets (e.g., ECoG, fMRI, MEG) to evaluate differences. In addition, I would love to emphasize the point of behavioral relevance: Not all the information that we can read out is used to guide behavior. It would be great if that could be considered in the discussion and evaluation of both models. In any case, an initial guideline on when to use linear versus nonlinear methods to analyze neuroimaging data will be very useful and beneficial to the CNN community and beyond. I am looking forward to the discussions and the outcome.

---

### Public Comment · ~Kristijan_Armeni1 · 2020-08-26
**When is the linear/non-linear distinction really crucial for the conclusions drawn?**

This is a timely proposal. Although linear models of brain activity have been present in cognitive neuroscience since the advent of cognitive neuroimaging [1] the use of ever more complex predictive models (often branded as “encoding” models) is gaining traction due availability of sophisticated (multivariate) statistical techniques and better availability of stimulus descriptions (“features”)  that operationalize cognitive hypotheses of interest. Overall, I think the workshop is well thought through and raises most of the important points in the field. I add some, hopefully useful and clear, comments pertaining to conceptual distinctions below.

**Models of data vs. models of the system**. It would be good to clarify the use of the term “model”. The distinction from [2] is good one to bear in mind, minimally: a) models of (neuroimaging) data (e.g. using regression to fit predictor X on brain data Y) and b) models of the system (e.g. task-performing spiking/artifical neural networks).
Although it is clear the proposal addresses a), it might be good to establish such a distinction, especially because if one is using e.g. artificial neural networks as (nonlinear) predictive models of brain data, the distinction (and resulting interpretations) can quickly become blurry and the term “model” overloaded.
In the light of Hypothesis 1 and 2. If we’re dealing with models of data, I dare say we’re likely not concerned with their “biological plausibility” as these are used as statistical tools linking our hypotheses (feature X) and our brain data (Y). In the same way that we are not too concerned with the biological plausibility of our signal processing tools on data Y.

**Prediction vs. interpretation vs. explanatory power**. This issue will pop up. As expressed by others, linear/non-linear distinction might be independent from predictive/explanatory goals. If I am modeling data and a linear model predicts data well, but a non-linear model predicts data even better (same feature, assuming fits are done correctly). Then by using linear models, I am missing out on the parts of variance not captured by the linear models. But my final interpretation of the fit still hinges on the choice of feature X (and perhaps comparison of several features), not so much on whether my linking model was linear/non-linear (as it is a model of data, not a direct model of the system).

**“Establish the theoretical validity of linear readout assumptions”**
I think, generally, this would be a good contribution. But I am not sure whether this assumption really affects the statements researchers make when fitting predictive (encoding) models? Say, my linear model $g(X)$ predicts $Y_a$ well, we’re saying some variance in $X$ goes with variance in $Y_a$ and we like to interpret this empirical fact saying that brain activity $Y_a$ “represents” $X$. To my understanding, we’re not concerned with how a downstream brain process $Y_b$ can linearly combine the process $Y_a$ to read out “information” $X$. Likely, this assumption is much more relevant when interpreting brain-brain (“connectivity”) linear models $g(Y_a) → Y_b$ or even when stimulus feature is included $g(Y_a | X) → Y_b$ (as in your 2.c.III use case).
Long story short, it might be worthwhile to be clear when the linear readout assumption is really problematic for conclusions drawn and whether/when the (non-) validity of the assumption can speak to the use of linear and non-linear models.

**Cognitive domain**. To make things more concrete, perhaps clearly specify if/which cognitive domain the GAC focuses on (vision, decision-making, language)? Or perhaps anticipate a systematic treatment for each.

**Benchmarking**. As a concrete outcome you mention an “online platform enabling researchers to systematically compare [...] models”. Perhaps as a provocation, do we want to establish a practice of “benchmarks” in neuroimaging, similar to the more applied machine learning fields, e.g. natural language processing (NLP)? Personally, I think it would be useful to know when my model is doing a decent job relative to some predefined criterion/past performances, before moving to interpretation. In the long run, I think we don’t want to embark on the benchmarking race.

All in all, excited to see the outcome of this GAC. I’m certain a lot of researchers will be interested in hearing more about the theoretical and practical considerations coming out of it.

References
[1]    K. J. Friston, A. P. Holmes, K. J. Worsley, J.-P. Poline, C. D. Frith, in R. S. J. Frackowiak, „Statistical parametric maps in functional imaging: A general linear approach“, Human Brain Mapping, let. 2, št. 4, str. 189–210, 1994, doi: 10.1002/hbm.460020402.
[2]    N. Kriegeskorte in P. K. Douglas, „Cognitive computational neuroscience“, Nature Neuroscience, let. 21, št. 9, str. 1148–1160, sep. 2018, doi: 10.1038/s41593-018-0210-5.

---

### Public Comment · ~Kohitij_Kar2 · 2020-08-26
**Its never that simple**

Overall comment:

I have really enjoyed reading this, and I think it is a very thought-provoking topic and might lead to a great debate from each side. Below are things that came to my mind as I went through the Abstract, and some are very nit-picky.

Let me first provide my current thoughts on the main scientific question posed by the authors: (I think I am on team nonlinear; but may be not fully) ..

As a description of brain activity, are linear models:
"accurate" —> by definition No.
"interpretable" —> depends on who is interpreting and how [I personally couldn’t care less].
"biologically plausible" —> these words are thrown around a lot these days and needs to be somehow operationalized. Can the authors synthesize a clear, quantifiable definition? That in itself will be worth the effort of having a debate.


My main suggestion to the authors is that they can improve the proposal by making the scope of this topic more explicit. For instance, there are so many types and forms of nonlinearity — and not all are equally relevant  for any problem (ranging from a  recurrent neural network to a  sigmoid). Are they allowing for any kind of nonlinear model? Or is there some kind of score on model complexity?

In the empirical branch, when the authors propose testing “mapping from stimulus to neural activity” or “neural activity to behavior”, “region to region”: my immediate thought was — how can these ever be linear? Its clearly nonlinear — the question is how much and is a linear model good enough for all current reasonable expectations out of neuroscience. So instead of linear vs. nonlinear, I think the primary debate could be how far away from linear transformations do we need to push the models.

I completely agree with the additional benefits of linear models (or as close as you can stay to linear models) that the authors mention — saves money, time, and computational resources.

Honestly, at the end of the debate, if we can make some progress towards quantifying the terms, “interpretable”, “biologically plausible” and “non-linear” — that will be great!

I am looking forward to seeing more of this!

---

### Public Comment · ~Anna_A_Ivanova1 · 2020-09-08
**Author Response to Reviewers (part 1)**

We thank all the reviewers for their detailed and insightful comments. Given the time constraints of the review response period, we have decided to write a single response addressing the main themes brought up in the reviews.

**1. No silver bullet**
 An important point raised by reviewers is that the desiderata for a model largely depend on the goals that model is built to achieve (Parthasarathy). In this proposal, we are concerned specifically with models designed for neuroscience research (with the goal to further our understanding of the brain) and not with models designed for engineering applications (e.g. brain-machine interfaces). In relation to this point, both prediction accuracy and interpretability are desirable properties of a model (as Emami pointed out). Therefore, we would attempt to benchmark models for both their prediction accuracy and interpretability, so that researchers can select models depending on their goals. As pointed out by Bashivan, Schwartz-Palleja,  Baldassano, Dobs, Gwilliams, and Rahnev, there likely isn’t one optimal model for all research applications: the goal of this project would be to offer guidance on what models might be most suitable for a specific goal. We will make clear that we do not expect either linear models (Hypothesis 1) or nonlinear models (Hypothesis 2) to be superior in all cases but are rather concerned with the factors that will make one or the other more suitable given the model objective.

**2. Prediction vs. interpretation**
Several reviewers pointed out that the linear/nonlinear debate is closely related to the prediction/interpretation debate (King, Hebart, Pramod). We agree that both the predictive power and interpretability should be considered when choosing between linear and nonlinear models. However, although the notion of “interpretability” is commonly used in the field and linear models are often considered more “interpretable”, Kar mentions that the notion of “interpretability” is not clearly defined. Indeed, there are multiple possible definitions of interpretability. A model could be considered “interpretable” if it has a small number of parameters or components. Alternatively, a model could be considered “interpretable” if its components can be concisely described with natural language (for example, a model of vision that represents object parts would be more “interpretable” than a model that represents functions of the input image that cannot be easily described), or if it can be derived from first principles or from a few simple equations. Finally, a model could be considered more “interpretable” if there is a “better” correspondence between components of the model and components of the brain (this latter notion of interpretability is related to the notion of biological plausibility). All these definitions of “interpretability” suffer from some degree of vagueness, and one of the goals of this proposal will be to develop benchmarks (as suggested by Armeni) to make at least some of these notions of interpretability quantifiable.

Pramod, Hebart, and Gwilliams suggested that linear models might be more “interpretable”. Given the same inputs, nonlinear models can often have more free parameters than linear models. However, as King pointed out, high-dimensional linear models might not be interpretable and, by contrast, nonlinear models could be interpretable when they are simple enough. Thus, we agree with the importance to distinguish between linearity and interpretability: in this project we will aim to develop both prediction accuracy and interpretability benchmarks for model evaluation, without posing a-priori that linear models are more interpretable.

**3. Linking brain and behavior**
Multiple reviewers pointed out the importance of connecting neural responses to behavioral outputs (Pramod, Dobs, Hebart, Kar, Parthasarathy). Many researchers operate under the assumption that if behavior is linearly decodable from a brain area, then that is good evidence for behavior actually being mechanistically read out of that area. However, especially with recent results showing that even dendrites can represent nonlinear functions, that assumption is coming into question, a point brought up by Hebart and Kar. Thus, Kar argued that, instead of a binary linear/non-linear discussion, we might have to figure out “how far away from linear transformations we need to push the models”. While we can build increasingly non-linear models connecting neurons to behavior, it will be interesting to discuss which experimental data are most suited to constrain this behavioral readout.

---

### Public Comment · ~Anna_A_Ivanova1 · 2020-09-08
**Author Response to Reviewers (part 2)**

**4. Limiting the scope - picking cognitive domains and neural recordings**
Many reviewers raised the valid point that there is unlikely to be a clear “winner” between linear and nonlinear models (Emami, Parthasarathy, Armeni, Pramod, Dobs, Bashivan, Baldassano, Schwartz-Palleja, and Rahnev) and asked us to specify which cognitive domains and/or which recording types we will consider (Toneva, Dobs, King, Parthasarathy, Armeni). We believe that, to provide a general response about the tradeoffs between linear and nonlinear models, it is important to cover multiple cognitive domains and examine multiple types of neural recordings. For the purposes of this proposal, we will focus on 3 cases:
a. Domain: vision. Data: human fMRI
b. Domain: vision. Data: single-neuron recordings in macaques
c. Domain: language. Data: human fMRI.
For all three cases, we will examine the link between the stimulus and the brain data. For b, we will additionally investigate the link between the brain data and behavioral output. These three cases should be sufficiently diverse for our conclusions to be generalizable. We encourage other researchers to join in and examine other domains (e.g. audition) and neuroimaging techniques (e.g. MEG).

**5. Limiting the scope - defining nonlinear models**
Kar and Pramod mention that the space of nonlinear models is very large, and it might therefore be possible to specify the types of nonlinearities we will consider. We operationalize the models as follows:
a. Linear: a ridge regression classifier that maps between X and Y, where X is a predictor of interest and Y is brain data.
b. Nonlinear: a neural network classifier with 1 or more ReLu layers that maps between X and Y, where X is a predictor of interest and Y is brain data.
Since neural networks are universal function approximators, we expect the nonlinear model to be able to capture a broad range of linking functions, at least in theory. The question of whether or not an appropriate function can be found in practice depends on the amount of training data and the number of layers and is one of the questions we intend to investigate.

**6. Connection to neural network interpretability debates**
Toneva mentions that the questions raised in our proposal are closely linked to recent work on neural network interpretability, particularly in NLP (natural language processing). Indeed, NLP researchers and neuroscientists are facing a similar problem: they are presented with a complex system (a neural net/the brain) and are trying to probe its contents. In fact, our proposal has been inspired by NLP discussions on this topic, including the Pimentel et al. paper that Toneva mentions. We look forward to exploring these parallels in more detail during the workshop (if our proposal is selected).

Overall, we are very excited by the level of interest in our proposal. We are happy to hear that many reviewers consider this to be a timely topic and look forward to further community discussions.